# Prognostic Value of Necroptosis-Related Genes Signature in Oral Squamous Cell Carcinoma

**DOI:** 10.3390/cancers15184539

**Published:** 2023-09-13

**Authors:** Ke Huang, Xiaoting Gu, Huimei Xu, Hui Li, Mingxuan Shi, Defang Wei, Shiqi Wang, Yao Li, Bin Liu, Yi Li

**Affiliations:** 1Key Laboratory of Preclinical Study for New Drugs of Gansu Province, School of Basic Medical Sciences, Lanzhou University, Lanzhou 730030, China; huangk18@lzu.edu.cn (K.H.); guxt21@lzu.edu.cn (X.G.); weidf21@lzu.edu.cn (D.W.); 2Key Laboratory of Dental Maxillofacial Reconstruction and Biological Intelligence Manufacturing, School of Stomatology, Lanzhou University, Lanzhou 730030, China; lih@lzu.edu.cn (H.L.); shimx2021@lzu.edu.cn (M.S.); wangshq2020@lzu.edu.cn (S.W.); 3Lanzhou University Second Hospital, Lanzhou University, Lanzhou 730030, China; xuhm18@126.com; 4The Third People’s Hospital of Gansu Province, Lanzhou 730030, China; lyao666999@gmail.com

**Keywords:** necroptosis, HPRT1, prognostic signature, biomarker, oral squamous cell carcinoma

## Abstract

**Simple Summary:**

This study aimed to explore the relationship between necroptosis and oral squamous cell carcinoma (OSCC) preliminarily, and search for the novel prognostic factors for OSCC, in order to help guide the mechanism exploration, clinical diagnosis, targeted therapy, and prognosis of OSCC patients. The prognostic signature of necroptosis-related genes (NRGs) for OSCC has been constructed through relevant bioinformatics methods, which showed that HPRT1 can serve as a novel independent prognostic factor for OSCC. The experiments also validated that the knockdown of HPRT1 could suppress the proliferation and migration of OSCC cells. Furthermore, the relationship between necroptosis and OSCC has also been discussed, which suggests that necroptosis might promote the development and metastasis of OSCC by initiating inflammatory response and immunosuppression.

**Abstract:**

The dual role of necroptosis in inhibiting and promoting tumor development has gradually received much attention because of its essential significance for targeted treatment. Accordingly, this study aims to explore the relationship between necroptosis and oral squamous cell carcinoma (OSCC), and search for novel prognostic factors for OSCC. RNA-seq data and clinical information were downloaded from TCGA and GTEx databases. The prognostic signature of necroptosis-related genes (NRGs) was constructed by univariate Cox regression analysis and the LASSO Cox regression model. Moreover, survival analyses, ROC curves, and nomograms were adopted to further analyze. GO and KEGG analyses and immune infiltration analyses were used for function enrichment and immune feature research in turn. The NRG prognostic signature expression was higher in OSCC tissues than in normal tissues, and the overall survival (OS) rate of the high-expression group was much lower. HPRT1 was proved to be an independent prognostic factor in OSCC. Furthermore, the function enrichment analyses revealed that NRGs were involved in necroptosis, apoptosis, inflammation, and immune reaction. The expression of NRGs was related to immunosuppression in OSCC. Furthermore, the knockdown of HPRT1 could suppress the proliferation and migration of OSCC. In conclusion, the high expression of NRG prognostic signature is associated with poor prognosis in OSCC, and HPRT1 can serve as a novel independent prognostic factor for OSCC.

## 1. Introduction

Oral squamous cell carcinoma (OSCC) is one of the most common head and neck malignancies with high incidence and mortality in certain parts of the world especially Asia [1,2,3,4,5,6,7,8,9]. Due to its potential for recurrence and metastasis, the 5-year survival rate of patients with OSCC has not been effectively improved currently [1,3,4,5,6,9,10,11]. Although some progress has been made with clinical treatments such as surgery, radiotherapy, chemotherapy, and targeted therapy, the overall treatment effect of OSCC has not been significantly improved because of its particular pathogenic location, restrictions, and the highly toxic side effects of traditional therapies [12,13]. Therefore, it is an urgent problem to explore new directions for effectively improving the diagnosis and prognosis of OSCC.

Necroptosis is a newly discovered form of programmed cell death (PCD) without dependence on caspase, which may play a promising role in the targeted therapy of tumors [14]. Many studies have suggested that necroptosis can act as an alternative way to inhibit tumor development when apoptosis is blocked. Still, some new studies have found that necroptosis can trigger inflammatory mechanisms to promote tumorigenesis and metastasis [15]. Inflammation is regarded as a typical hallmark of cancer that can promote tumor development and progression and may be associated with a poor prognosis [16]. Liu et al. proposed that the knockdown of RIPK1, RIPK3, and MLKL in breast cancer cells can reduce the production of proinflammatory cytokines, thereby reducing tumorigenicity [17]. Strilic et al. suggested that RIPK1 inhibitor and deletion of RIPK3 reduced the necroptosis of endothelial cells induced by lung cancer cells and inhibited tumor cell extravasation and metastasis [18]. It has been shown that downregulation of RIP3 in pancreatic ductal adenocarcinoma (PDA) can increase lymphocyte infiltration and eliminate immune tolerance so that the necroptosis may promote the occurrence of PDA through inducing immunosuppression [19]. Moreover, the auxo-action of necroptosis in esophageal cancer, intestinal cancer, melanoma, and so on has also been reported [17,20,21,22].

Currently, necroptosis has been shown to be a double-edged sword in cancer initiation and progression [2,14,23], as well as OSCC. On the one hand, triggering necroptosis could be an alternative pathway to overcome apoptotic resistance in cancer treatments [2,6,24]. It has been reported that certain phytochemicals and compounds can enhance the sensitivity of OSCC to conventional anticancer drugs and radiation therapy by inducing necroptosis [2,25]. On the other hand, it was also found that suppressing the mechanism of necroptosis can induce apoptosis in OSCC. Xu et al. suggested that knockdown of RIP1 could increase the oxaliplatin-induced apoptosis in human OSCC [26]. Yun et al. proposed that the inhibitory effects of Mach against OSCC are related to the promotion of apoptosis and inhibition of necroptosis [9]. Overall, as with nonapoptotic cell death, there are still relatively few reports about necroptotic mechanisms in the treatment of OSCC [2]. It is necessary to conduct a more in-depth and thorough exploration to confirm the antitumoral or protumoral properties of necroptosis in OSCC.

In this study, we use bioinformatics to retrieve necroptosis-related genes with a prognostic role in OSCC through relevant databases, and verify the selected gene by proliferation and migration assays in vitro, so as to provide the novel prognostic biomarkers regarding necroptosis for the targeted therapy in OSCC.

## 2. Materials and Methods

### 2.1. Data Collection and Integration

In this study, the gene expression profiles consisted of 329 OSCC samples from TCGA (https://portal.gdc.cancer.gov/) (accessed on 5 October 2022) and 32 normal samples from GTEx for identifying prognosis-related genes. RNA-seq data along with the associated clinical data were obtained from the TCGA-HNSC project in level 3 HTSeq-FPKM format. Subsequently, these data were transformed into the transcripts per million reads (TPM) format. Samples from sites associated with OSCC (alveolar ridge, base of tongue, buccal mucosa, floor of mouth, hard palate, oral cavity, oral tongue) were included in the study, while those from non-OSCC sites (hypopharynx, larynx, lip, oropharynx, tonsil) or those with unclear or incorrect information were excluded to ensure reliable results. Supplementary prognostic data were obtained from a study on Cell [27]. The necroptosis-related gene set was obtained from the GeneCards database (https://www.genecards.org) (accessed on 5 October 2022). Genes meeting the criteria of hazard ratios (HR) > 1, 95% confidence intervals (CI) > 0.4, and *p* < 0.05 were regarded as genes associated with necroptosis. 

### 2.2. Construction of the NRGs Prognostic Signature

To start with, the log2-fold change (log2FC) was calculated to compare mRNA expression levels between tumor and normal samples. Subsequently, the samples in each group were segregated into high- and low-expression groups based on the median expression levels of the target genes. The genes within the intersection of the TCGA and GeneCards datasets, which were previously extracted, underwent univariate Cox regression analysis via the R package (survival, version 3.2.10) to estimate the prognostic value. The genes with significant prognostic value (*p* < 0.05) were filtered to construct the least absolute shrinkage and selection operator (LASSO) regression by the R package “glmnet” (version 4.1.2) and “survival” (version 3.2.10), which can prevent information overfitting. According to the median risk score, the samples enrolled were split into low-risk and high-risk groups. Next, a visual analysis of the previously identified prognostic-related NRGs was conducted using risk curves, scatter plots, and heatmaps (R package “ggplot2”, version 3.3.3).

### 2.3. Construction of KM Survival Curves, ROC Curves, and Nomograms

The Kaplan–Meier (KM) survival curves were generated using the R packages “survminer” (version 0.4.9) and “survival” (version 3.2.10). The time-dependent receiver operating characteristic (ROC) curves were employed for assessing the prediction model’s accuracy using the R packages “timeROC” (version 0.4) and “ggplot2” (version 3.3.3). The calculation of the area under the curve (AUC) was performed to summarize the model’s performance. Nomograms and calibration curves were constructed using the R packages “rms” (version 6.2.0) and “survival” (version 3.2.10). Nomograms were utilized for predicting patient survival probabilities based on multiple variables, while calibration curves were employed to assess the accuracy of the predictive nomogram model.

### 2.4. Multivariate Cox Regression Analysis and Differential Expression Analysis 

Multivariate Cox regression analyses were constructed to assess whether each gene in the constructed model can function as an independent prognostic factor. The R package “survival” (version 3.2.10) was employed for data processing. Genes with Hazard Ratios (HR) greater than 1 and *p*-values less than 0.05 were considered independent prognostic factors. Additionally, the differential expressions of six NRGs between patients with OSCC and normal samples were investigated (utilizing the R package “ggplot2”, version 3.3.3). These findings were further validated by the interrelated immunohistochemical information of OSCC patients from the Human Protein Atlas (HPA) database. 

### 2.5. Pathway and Function Enrichment Analyses

For the purpose of investigating the potential biological functions and pathways correlated with the expression of prognostic NRGs, Gene Ontology (GO), and Kyoto Encyclopedia of Genes and Genomes (KEGG) analyses were conducted using the R package clusterProfile (version 3.14.3) and org.Hs.eg.db (version 3.10.0) [28]. At the same time, protein–protein interaction (PPI) network analyses via the STRING database (https://cn.string-db.org/) (accessed on 5 October 2022) and the GeneMANIA (http://genemania.org) (accessed on 5 October 2022) were used for finding relevant proteins related to NRGs [29]. Additionally, we used Gene correlation analysis to construct the interactive network using Spearman correlation with the R package “stat” (version 3.6.3). The data with *p* value < 0.05 were considered statistically significant.

### 2.6. Immune Cell Infiltration Analysis 

The relationship between the expression of NRG prognostic signature and immune cell infiltration across 24 distinct immune cell types within OSCC was investigated. These immune cell types included aDC (activated DC), B cells, CD8 T cells, Cytotoxic cells, DC, Eosinophils, iDC (immature DC), Macrophages, Mast cells, Neutrophils, NK CD56 bright cells, NK CD56 dim cells, NK cells, pDC (Plasmacytoid DC), T cells, T helper cells, T central memory (Tcm), T effector memory (Tem), T follicular helper (Tfh), T gamma delta (Tgd), Th1 cells, Th17 cells, Th2 cells, and Treg. The markers for the 24 immune cell types were derived from a study by Bindea et al. [30]. To conduct the immune cell infiltration analysis, we employed the R package GSVA package (version 1.34.0) developed by Hänzelmann et al. [31]. Specifically, we utilized the ssGSEA algorithm, an integrated method within the GSVA package, which allowed us to quantify the enrichment levels of immune cell signatures within each OSCC sample.

### 2.7. Cell Culture

The human OSCC cell lines SAS was purchased from Cellcook (Guangzhou, China), and SCC-9 was from Procell (Wuhan, China). The cells were cultured in Dulbecco’s Modified Eagle Medium/Nutrient Mixture F-12 (DMEM/F-12) (BasalMedia, Shanghai, China), supplemented with 10% fetal bovine serum (FBS) (ABW, Shanghai, China), and Penicillin (100 U/mL), Streptomycin (0.1 mg/mL), Gentamicin (50 μg/mL) Solution (Solarbio, Beijing, China). In addition, the growth factor Hydrocortisone (400 ng/mL) was added to the medium of SCC-9. Cells were cultured in a humidified incubator and maintained at 37 °C and 5% CO_2_. 

### 2.8. Cell Transfection

The HPRT1 was selected as the representative gene for functional validation in OSCC cell lines. Cells were cultured in a flask until they reached an appropriate density and exhibited good cellular condition. Subsequently, small interfering RNA (siRNA) transfection was performed to knock down the HPRT1 expression using Opti-MEM (Gibco, Thermo Fisher, USA) containing the siRNA and 80 nM Lipofectamin 2000 Reagent (Invitrogen, Thermo Fisher, USA). The siRNAs were obtained from Genepharma (Shanghai, China). After transfection for 6 h, the medium was replaced with a complete medium. Once the cell state stabilized, cells were used for subsequent experiments. The corresponding siRNA sequences are as follows: SiRNA-NC: sense: 5′-UUCUCCGAACGUGUCACGUTT-3′; antisense: 5′-ACGUGACACGUU CGGAGAATT-3′. SiRNA-1: sense: 5′-GCCCUUGACUAUAAUGAAUTT-3′; antisense: 5′-AUUCAUUA UAGUCAAGGGCTT-3′. SiRNA-2: sense: 5′-GGCAGUAUAAUCCAAAGAUTT-3′; antisense: 5′-AUCUUUGG AUUAUACUGCCTT-3′. SiRNA-3: sense: 5′-GAUGAUCUCUCAACUUUAATT-3′; antisense: 5′-UUAAAGUU GAGAGAUCAUCTT-3′. SiRNA-4: sense: 5′-CAUACCUAAUCAUUAUGCUTT-3′; antisense: 5′-AGCAUAAU GAUUAGGUAUGTT-3′.

### 2.9. RNA Extraction and RT-qPCR

The efficiency of gene knockdown was assessed through RT-qPCR. Total RNA was extracted following the manufacturer’s instructions using the SPARKeasy Cell RNA Kit (Sparkjade, Jinan, China). The RNA was then reverse transcribed into cDNA using a two-step reverse transcription reaction, followed by qPCR using the Evo M-MLVRT Mix Kit Ver.2 (AG, Changsha, China) as per the recommended procedure. The primers for GAPDH and HPRT1 were obtained from Servicebio (Wuhan, China). The corresponding primer sequences are as follows: GAPDH: sense: 5′-GGAAGCTTGTCATCAATGGAAATC-3′; antisense: 5′-TGATGACC CTTTTGGCTCCC-3′. HPRT1: sense: 5′-CCTGGCGTCGTGATTAGTGAT-3′; antisense: 5′-AGACGTTCAGTCCTGTCCATAA-3′.

### 2.10. Cell Growth and Cell Proliferation Assay

Cell growth rates were assessed using the commercial Cell Counting Kit-8 (CCK-8) from Abmole (Houston, TX, USA). Cells were seeded at a density of 2000 cells per well in 96-well plates. Upon completion of seeding, CCK-8 was immediately added, and the cells were incubated for 2 h before measuring the absorbance. Additionally, after culturing for 1, 2, and 3 days, CCK-8 was again added and incubated for 2 h before absorbance measurement. The optical density (OD) values were quantified using a microplate reader (Infinite M200 Pro, Tecan Group, Männedorf, Switzerland) at a wavelength of 450 nm. The formula for calculating the relative cell growth rate is as follows: Relative cell growth rate = OD_(Day 0)_/OD_(Day 1/2/3)_ × 100%.

### 2.11. Wound Healing Assay

Wound healing assay was performed to assess the cell migration ability. Cells were seeded into 6-well plates and allowed to reach approximately 90% confluency. Once the cells reached a healthy growth state and appropriate density, cell transfection was conducted. After transfection, a straight scratch wound was carefully created across the cell monolayer using a sterile 10 μL pipette tip. After creating the wound, the cells were gently washed with phosphate-buffered saline (PBS) to remove cell debris. Subsequently, fresh serum-free cell culture medium was added to the wells, and the cells were allowed to migrate and close the wound for a specified period. The specific duration varies depending on the cell type. Microscopy was used to capture images of the initial wound and the wound closure at the end of the incubation period. 

### 2.12. Transwell Assay

Transwell assay was employed to evaluate the cell migration capability. In brief, Transwell inserts with 8.0 μm pore size membranes (Corning Incorporated, Corning, NY, USA) were placed into 24-well plates. For the migration assay, the upper chambers were filled with serum-free cell culture medium, and cells were seeded into the upper chambers at a density of 10^5^ cells per well. The lower chambers contained medium supplemented with FBS. The cells were allowed to migrate through the membrane for 24 h. After incubation, non-migratory or non-invasive cells on the upper surface of the membrane were gently removed with a cotton swab, while those that had migrated or invaded were fixed and stained with 0.1% crystal violet. The number of migrated cells was counted under a microscope in multiple randomly selected fields. 

### 2.13. Statistical Analyses and Data Visualization

The bioinformatic-related statistical calculations and data visualizations were conducted using R software (version 3.6.3, R Core Team, Vienna, Austria). For statistical analysis of the experimental data in the functional validation part, one-way ANOVA or two-way ANOVA was performed using the GraphPad software (version 8.3.0, San Diego, CA, USA). The final assembly of figures was performed using Adobe Illustrator software (version 25.0.0.60, Adobe Inc., Mountain View, CA, USA).

## 3. Results

### 3.1. Screening of Prognostic NRGs

The flowchart for the bioinformatics research methodology is illustrated in Figure 1. A total of 329 OSCC samples were sourced from TCGA, accompanied by the relevant clinical characteristics such as age, gender, and clinical stage (stage I, stage II, stage III, and stage IV) (Table 1). As depicted in the Venn diagram, there were 2245 prognosis-related genes identified from OSCC patients in the TCGA database and 195 necroptosis-related genes sourced from the GeneCards database. Only 26 genes were included in the intersection of both datasets (Figure 2). Specific gene lists have been provided in Appendix A. Univariate Cox regression analysis was performed, which further demonstrated the HR of the 26 screened genes (Figure 3). 

### 3.2. Construction of LASSO Cox Regression Model 

The 26 prognostic NRGs, identified through univariate Cox regression, were subsequently subjected to a least absolute shrinkage and selection operator (LASSO) regression model for dimension reduction. Ultimately, six genes (HPRT1, PGAM5, BID, SMN1, FADD, and KIAA1191) were selected to construct the prognostic signature for NRGs (Figure 4A,B). Subsequently, we employed the risk curves, scatter plots, and heat maps of the resulting genes to characterize the LASSO Cox regression model. These analyses revealed that the high-risk group of OSCC patients exhibited elevated risk scores and mortality rates compared to the low-risk group. Furthermore, the six genes comprising the NRGs prognostic signature demonstrated higher expression levels in cancer tissues than in normal tissues (Figure 4C–E). The information presented above serves as validation for the rationality of the previously constructed risk score model.

### 3.3. Validation of NRGs Prognostic Signature 

The KM survival curves, ROC curves, nomograms, and calibration curves were generated to further identify the features of the six prominent genes in the NRG prognostic signature. The KM survival curves showed that the overall survival (OS) rate of the high-risk group was significantly lower than the low-risk group, indicating a notable positive correlation between the high expression level of NRGs and the risk scores (Figure 4F). The AUC of the ROC curve was used to assess the diagnostic value of NRGs at 1, 3, and 5 years for OSCC patients. The AUC values, in ascending order, were 0.597, 0.652, and 0.674, indicating that the diagnostic efficacy increased with longer times (Figure 4G). Subsequently, the clinical characteristics of the constructed signature were presented visually through the nomogram, incorporating variables such as age, gender, clinical stage, HPRT1, PGAM5, BID, SMN1, FADD, and KIAA1191. The calibration curves confirmed the reliability of the nomogram. Based on the results, it can be inferred that HPRT1, age (greater than 60), BID, and PGAM5 might be relatively influential risk factors in the prognosis of OSCC (Figure 5).

### 3.4. Independent Prognostic Value and Expression of the NRGs Signature 

To further delve into the prognostic signature for NRGs, the six genes were included in the multivariate Cox regression analyses along with other significant clinical factors. Using multivariate Cox regression, it was observed that only HPRT1 and BID could function as the independent prognostic factors among the six NRGs (HR > 1 and *p* ≤ 0.05). This result suggests that HPRT1 and BID could independently influence the prognosis of OSCC patients, separate from the prognostic model composed of the six genes in this study. Additionally, among the six genes, HPRT1 may have the most substantial adverse impact on the prognosis of OSCC (Figure 6). Differential expression analyses were conducted to compare the expression levels of the 6 NRGs in OSCC and normal tissues. The results showed that KIAA1191 exhibited no significant difference in expression between OSCC and normal tissues. Meanwhile, HPRT1, PGAM5, BID, SMN1, and FADD exhibited higher expression in OSCC tissues (Figure 7A,B). 

In addition, the immunohistochemical information of 6 NRGs was downloaded from the HPA database to verify the differential expression analyses qualitatively (Figure 7C–N). Higher-resolution original images can be found in the Appendix A. Given the absence of OSCC-related samples in the HPA database, we included the head and neck squamous cell carcinoma (HNSC) samples and their corresponding normal tissue samples for comparative analysis. Specifically, from the obtained immunohistochemical results, in normal tissues (Figure 7C–H), the staining intensity of the target genes was observed as negative (BID, HPRT1), weak (FADD), or medium (KIAA1191, PGAM5, SMN1). In tumor tissues (Figure 7I–N), the staining intensity of the target genes was strong (BID, HPRT1, SMN1), high (PGAM5), or medium (FADD, KIAA1191). These results corroborate the earlier findings. Following that, KM survival curve analyses were employed to depict the relationship between OS and the expression levels of the 6 NRGs. The results of the KM curves indicated that, among all the genes, high expression of each gene is significantly associated with poorer survival probability. (Figure 7O–T). Furthermore, ROC curves were utilized to calculate the predictive power in terms of sensitivity (TPR) and specificity (FPR) for the six genes. The results reveal that, among the six genes, all except for KIAA1191 exhibit good predictive power (Figure 7U–Z).

### 3.5. Functional Enrichment Analyses of Prognostic NRGs

The known, predicted, and other interactions between prognostic NRGs and the related proteins were plotted using the STRING database (Figure 8A). The networks among 26 NRGs were visualized by gene correlation analyses with the associated functions, such as extrinsic apoptotic signaling pathway, positive regulation of peptidase activity, positive regulation of proteolysis, positive regulation of apoptotic process, programmed necrotic cell death, apoptotic mitochondrial changes, and necrotic cell death (data collected from GeneMANIA) (Figure 8B). GO and KEGG analyses were employed to explore the potential biological functions of genes associated with the expression of NRGs, such as apoptosis, necroptosis, TNF signaling pathway, cytokine receptor binding, tumor necrosis factor receptor superfamily binding, tumor necrosis factor receptor binding, death receptor binding, positive regulation of proteolysis, extrinsic apoptotic signaling pathway, positive regulation of peptidase activity, and I-kappaB kinase/NF-kappaB signaling (Figure 8C,D). The detailed analytical data are shown in Appendix A.

### 3.6. Immune Cell Infiltration Analysis of NRGs

Considering that some existing studies have shown that necroptosis is associated with immunosuppression, immune cell infiltration analysis was further performed. Combining the previous investigations and immune infiltration analysis, we found that the independent prognostic factors of HPRT1 and BID showed a significantly negative correlation with immune cell infiltration (Figure 9A,B). In addition, the other genes of NRG prognostic signature (except for KIAA1191) were also found to be negatively associated with immune infiltration (Figure 9C–F).

### 3.7. Knockdown of HPRT1 Attenuated the Growth and Migration Capabilities of OSCC Cell Lines

Previous analyses indicated that HPRT1 is the most representative oncogene for the prognosis-related NRGs in OSCC. Therefore, to establish the correlation between HPRT1 and several representative hallmarks of OSCC, HPRT1 was knocked down in SAS and SCC-9 cells using siRNA transfection technology. RT-qPCR was conducted to confirm the gene knockdown effects (Figure 10A). Next, a CCK-8 assay was used to measure the tumor growth rate after gene knockdown. The results showed that on the first day after transfection, both cell lines’ growth rates did not exhibit significant changes compared to non-transfected cells (negative control, NC). The decrease in cell growth rate became gradually evident on the second day and significantly decreased on the third day. Overall, the relative cell growth rate indicated that both SAS and SCC-9 cells grew slower when HPRT1 was knocked down (Figure 10B). In the meantime, the results of the wound healing assay and Transwell assay indicated a decrease in migration ability when HPRT1 was knocked down (Figure 10C,D).

## 4. Discussion

Currently, the mortality and morbidity of OSCC patients remain high, which is still a growing global health concern [1,2,32]. The roles of necroptosis in various types of tumors have been reported, but there is limited information regarding the treatment of OSCC through the necroptotic mechanism [2].

In this study, the significance of certain genes as prognostic indicators for OSCC was explored. Notably, HPRT1 emerged as a robust independent prognostic indicator. HPRT1, known for its involvement in purine and inosine synthesis, as well as its role in cell cycle regulation and DNA replication [33], exhibited increased expression in various malignancies, including breast, colon, and rectal cancers [34,35,36]. Recent research linked elevated HPRT1 levels in OSCC to poor prognosis, chemoresistance, and activation of the MMP1/PI3K/Akt signaling pathway [37]. Similarly, BID, a proapoptotic member of the BCL-2 family, stood out in the OSCC prognostic signature [38]. While apoptosis-related factors are often associated with tumor suppression [14], BID’s role in promoting tumor progression was highlighted, potentially through non-apoptotic mechanisms [38,39,40]. Further investigations indicated BID’s involvement in promoting hepatocarcinogenesis through cell proliferation and inflammation activation [41,42,43]. Another gene of interest, PGAM5, a mitochondrial phosphatase, demonstrated elevated expression in various cancers and a potential role in initiating necroptosis [44,45,46,47]. FADD and KIAA family-related genes were also implicated in tumor progression [48,49,50], while SMN1 showed connections to immune function and poor prognosis in related tumors [51,52]. Collectively, these findings suggest that these genes contribute to OSCC development, metastasis, and immune modulation, impacting prognosis adversely.

In summary, the NRGs play a significant role in the development and prognosis of OSCC. Combined with the results of bioinformatics and experiments in vitro, we speculate that the mechanism of necroptosis in OSCC may be as follows: On the one hand, TNFα binds to TNFR in the cell membrane and successively recruits to form complex I, complex II, and necrosome. After phosphorylation of MLKL, PGAM5, and DRP1 are activated, leading to mitochondrial fission and necroptosis [17]. Meanwhile, the pro-apoptotic gene BID can activate the growth-orientated pathways by promoting both cell proliferation and compensatory stimulus of tumor [42]. On the other hand, a large number of cytokines, cell contents, and DAMPs released by necroptosis may trigger the NF-kappaB pathway and initiate and activate the NLRP3 inflammasome, which can cause a robust inflammatory response and immunosuppression to promote the tumor growth and metastasis [53,54,55]. A detailed mechanistic diagram is shown in Figure 11. 

There are still some deficiencies in our study. In the future, we will continue to work on further validating HPRT1 in OSCC with cell lines, animal models, and human tissue. Moreover, our speculation on the mechanism of necroptosis in the development and prognosis of OSCC requires further verification by related experiments. Searching for the potential drugs targeting HPRT1 for OSCC treatment is also in our plan.

## 5. Conclusions

In conclusion, we constructed an NRG signature in this study, which could help to direct the clinical diagnosis and prognosis of OSCC patients as new biomarkers. As certified through the preliminary study, the high expression of HPRT1 could serve as a poor prognostic factor independently in OSCC. 

## Figures and Tables

**Figure 1 cancers-15-04539-f001:**
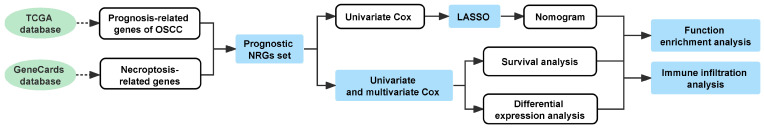
Flowchart for the bioinformatics research methodology.

**Figure 2 cancers-15-04539-f002:**
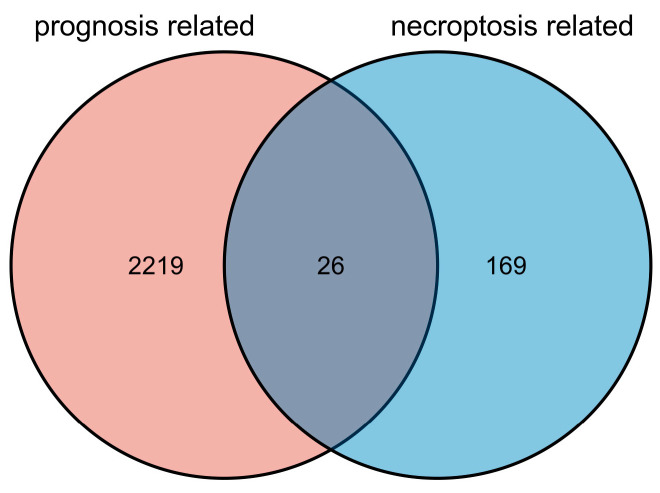
Venn diagram of co-expressed genes from two databases.

**Figure 3 cancers-15-04539-f003:**
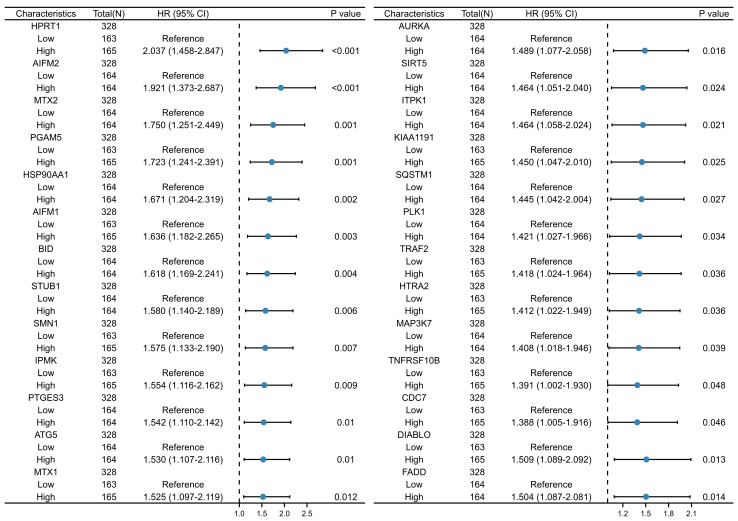
Univariate Cox regression analyses of the prognostic NRGs.

**Figure 4 cancers-15-04539-f004:**
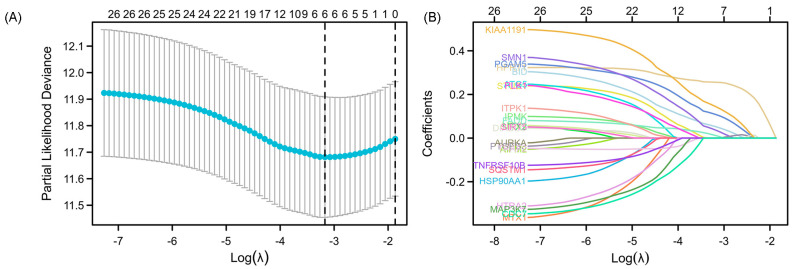
Selection and validation of NRGs prognostic signature with LASSO. (**A**) LASSO cross-validation plots of prognostic NRGs. (**B**) LASSO coefficients of prognostic NRGs. (**C**) Risk curves of high-risk and low-risk groups with OSCC based on NRGs. (**D**) Scatter plots comparing the survival time of OSCC patients with alive status to dead ones based on NRGs. (**E**) Heatmap showing the high expression of NRGs prognostic signature in OSCC contrary to normal samples. Red represents upregulated NRGs, blue represents downregulated NRGs, and white represents no significant changes. (**F**) KM survival curve analysis. (**G**) ROC curves based on NRGs at 1, 3, and 5 years.

**Figure 5 cancers-15-04539-f005:**
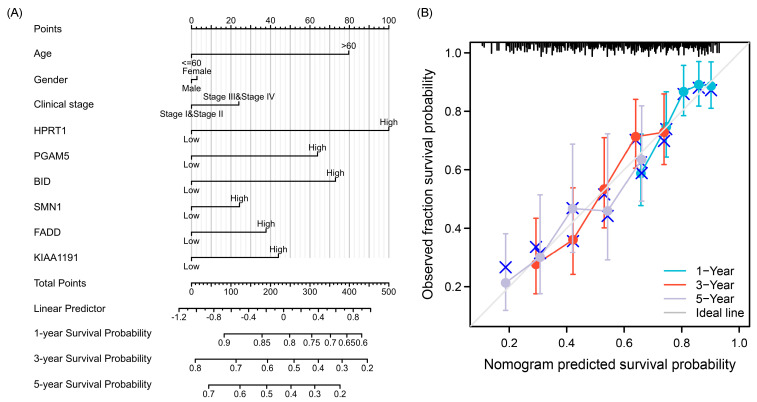
(**A**) The nomogram based on the expression of prognostic NRGs with vital clinical characteristics. (**B**) The calibration curve of the nomogram to predict the survival probability at 1, 3, and 5 years.

**Figure 6 cancers-15-04539-f006:**
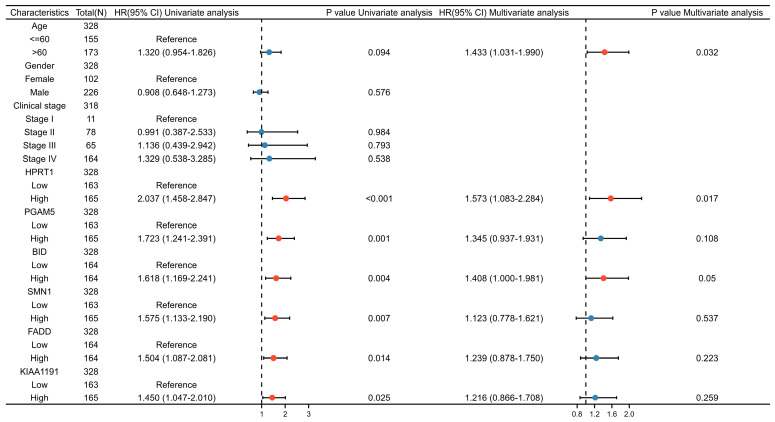
Univariate and multivariate Cox regression analyses of six prognostic NRGs and clinical factors.

**Figure 7 cancers-15-04539-f007:**
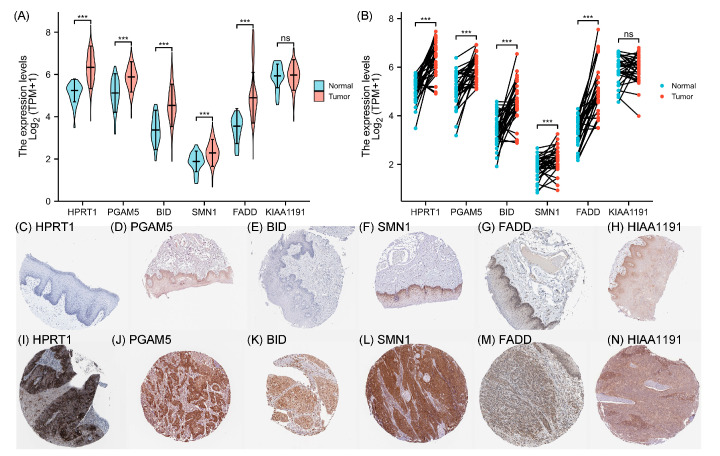
The expressive levels and survival analyses of 6 NRGs in OSCC. (**A**) Differential expression analyses of non-paired samples comparing the expressive levels of 6 NRGs in OSCC and normal tissues. (**B**) Differential expression analyses of paired samples comparing the expressive levels of 6 NRGs in OSCC and normal tissues. (**C**–**H**) The expressive levels of HPRT1, PGAM5, BID, SMN1, FADD, and KIAA1191 in normal tissues from HPA. (**I**–**N**) The expressive levels of HPRT1, PGAM5, BID, SMN1, FADD, and KIAA1191 in OSCC patients from HPA. (**O**–**T**) The KM survival curves describe associations between overall survival and expressions of 6 NRGs. (**U**–**Z**) The ROC curves of the 6 NRGs. (ns, no significance; ***, *p* < 0.001).

**Figure 8 cancers-15-04539-f008:**
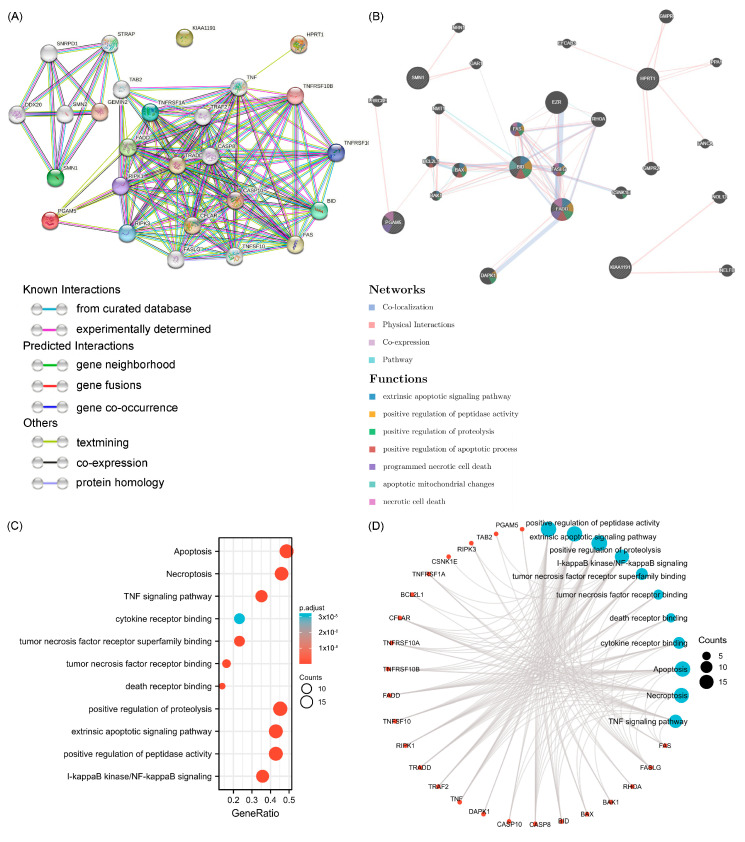
Correlation interactive networks and GO/KEGG enrichment analyses of NRGs. (**A**) Protein-protein interaction (PPI) networks of 26 proteins interrelated with NRGs. (**B**) Network diagram of 26 genes correlated with NRGs. (**C**) Bubble plot of GO and KEGG enrichment analyses. (**D**) Network visualization of 24 NRGs with 11 potential biological functions and pathways.

**Figure 9 cancers-15-04539-f009:**
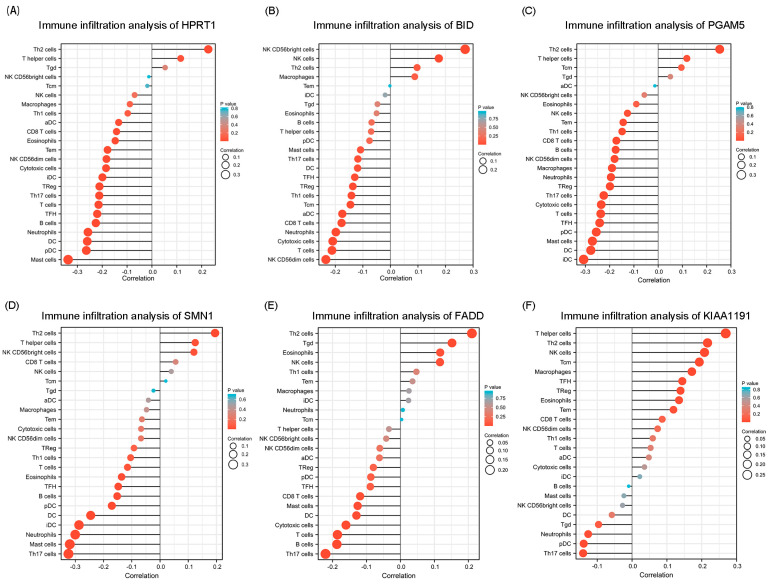
Immune cell infiltration analyses of prognostic NRGs. (**A**) Immune infiltration analysis of HPRT1. (**B**) Immune infiltration analysis of BID. (**C**) Immune infiltration analysis of PGAM5. (**D**) Immune infiltration analysis of SMN1. (**E**) Immune infiltration analysis of FADD. (**F**) Immune infiltration analysis of KIAA1191.

**Figure 10 cancers-15-04539-f010:**
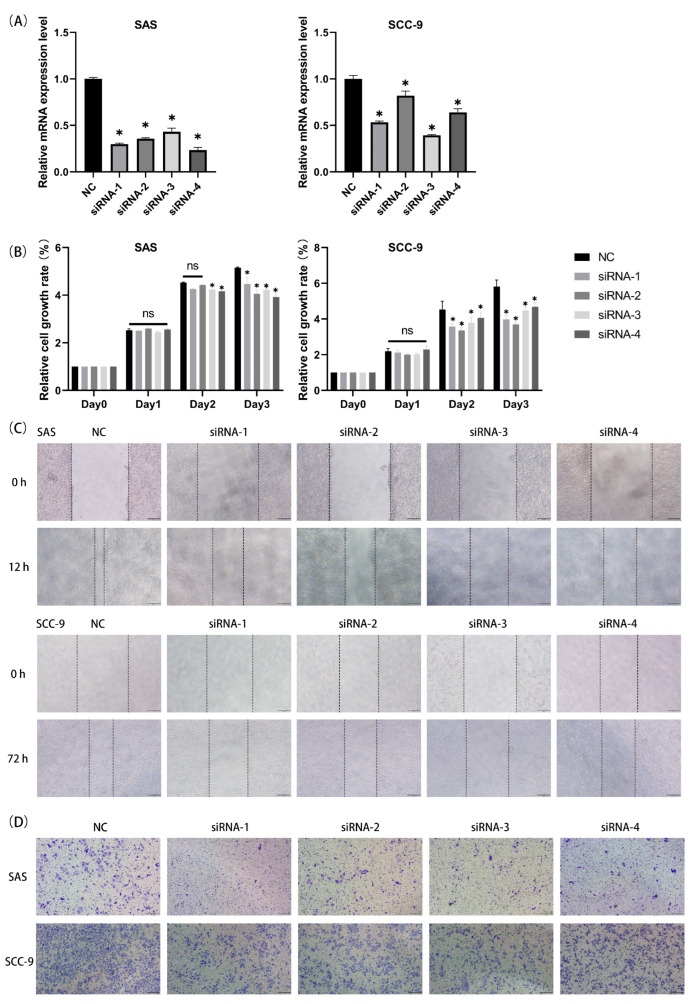
Knockdown of HPRT1 attenuated the growth and migration capabilities of OSCC cell lines. (**A**) RT-qPCR demonstrated excellent knockdown efficiency of HPRT1 in both cell lines. (**B**) The relative cell growth rate was assessed using the CCK-8 assay. (**C**) Cell migration ability was evaluated using the wound healing assay. (**D**) Cell migration ability was assessed using the Transwell assay. (ns, no significance; *, *p* < 0.05).

**Figure 11 cancers-15-04539-f011:**
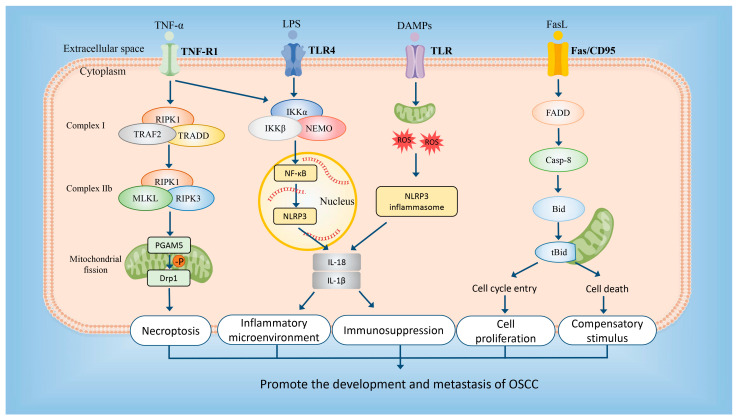
The related mechanistic diagram based on prognostic NRGs in OSCC. The prognostic NRGs may promote the development and metastasis of OSCC by participating in the process of necroptosis, cell proliferation, compensatory stimulus, inflammatory microenvironment, and immunosuppression.

**Table 1 cancers-15-04539-t001:** Baseline information for the OSCC samples obtained.

Characteristic	Levels	Overall
N		329
Clinical stage, n (%)	Stage I	11 (3.4%)
	Stage II	79 (24.8%)
	Stage III	65 (20.4%)
	Stage IV	164 (51.4%)
Age, n (%)	≤60	155 (47.3%)
	>60	173 (52.7%)
Gender, n (%)	Female	102 (31%)
	Male	227 (69%)
Age, median (IQR)		61 (54, 70.25)

## Data Availability

The data presented in this study are available in the article and Appendix A.

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
