# Peer review of "Prognostic Value of Necroptosis-Related Genes Signature in Oral Squamous Cell Carcinoma"

_cancers, 2023, doi:10.3390/cancers15184539_

Round 1
Reviewer 1 Report
The manuscript reports the prognostic value of necroptosis-related genes signature in oral squamous cell carcinoma (OSCC). The study is interesting. However, some revisions are needed. My comments are as follows.
Major Revisions
#1:
All assays seem to be in silico data analyses. Basically, the assays are easy to analyse. Therefore, HPRT1 should be validated with cell lines, animal models, or human tissue of the authors' institution.
#2: lines 137–:
The method of the immune cell infiltration analysis seems to be ambiguous. Please clarify the details how to analyse.
#3: lines 68–70:
The authors claim that necroptosis's impact on OSCC is still unclear. However, I don't think so. Some researches have already reported the necroptosis on OSCC. For example,
- J Stomatol Oral Maxillofac Surg. 2023 Jul 15:101565. doi: 10.1016/j.jormas.2023.101565.
- Indian J Clin Biochem. 2023 Jul;38(3):351-360. doi: 10.1007/s12291-022-01055-7.
- Front Pharmacol. 2021 Aug 27;12:676813. doi: 10.3389/fphar.2021.676813.
- Oncotarget. 2016 Aug 5;8(36):60060-60079. doi: 10.18632/oncotarget.11085.
- Oncol Rev. 2018 Mar 27;12(1):358. doi: 10.4081/oncol.2018.358.
The above articles were easily found with PubMed and Google Scholar. So, I think the authors intentionally hide those articles to appeal their study's novelty. This makes the study very unreliable. To higher the novelty of the study, a rigid literature review must be written, and state “what is known, and what is unknown”.
#4: lines 261– (Discussion section):
The section is too long. Please shorten. Some sentenses are overlapped with the other sections.
Minor Revisions
#5: lines 43–44:
I don't think the references 1 and 2 adequately report that 'OSCC is one of the most common head and neck malignancies, and its incidence has increased year by year in the world'. So please change the references.
#6: lines 44–46:
Like the comment #4, the references 2 and 3 might be changed them to the more adequate ones.
#7:
I don't think OSCC is the sixth most common malignancy in the world. Please confirm.
Comment #8: line 121:
Please correct OCSS to OSCC.
Reviewer 2 Report
The article submitted by the Huang et al., explore the role of Necroptosis related gene signature in OSCC. Though the author's experimental plan and validation is good. I have several concerns
1) Authors used the regression models
a) Univariate regression model
b) After filtering the gene set they used the Multivariate regression model
But authors did not explain which genes were best correlated with the patient clinical parameter.
2) Further, after using the K-M survival analysis, the authors did not calculate the prognostic difference between the survivals.
3) Additionally the authors did not calculate the predicting power (sensitivity & specificity ) of the prognostic gene signature.
4) The result of the Cox regression model was not validated by using an independent dataset.
Without the above analysis, the MS is not acceptable. I recommend a major revision for this article.
Reviewer 3 Report
Your article has certainly captured significant attention due to its vital implications for targeted treatment. I commend your efforts in delving into the intricate relationship between necroptosis and oral squamous cell carcinoma (OSCC), seeking novel prognostic factors for OSCC. Your utilization of RNA-seq data from TCGA and GTEx databases to construct the necroptosis-related prognostic signature is commendable, employing both univariate Cox regression and LASSO Cox regression analyses. The integration of survival analysis, ROC curves, and nomograms further enriches the study. Notably, your comprehensive exploration through GO, KEGG analyses, and immune infiltration investigation enhances our understanding of functional enrichment and immune features. The revelation of higher NRG expression in OSCC tissues, correlating with poorer overall survival, underscores the clinical significance. The identification of HPRT1 as an independent prognostic factor in OSCC is an insightful addition. Your enrichment analyses illuminating NRG involvement in necroptosis, apoptosis, inflammation, and immune response, as well as its correlation with immunosuppression in OSCC, paints a comprehensive picture. In conclusion, your study provides compelling evidence for the adverse prognosis associated with elevated NRG prognostic signature expression in OSCC, and the newfound role of HPRT1 as an independent prognostic factor is promising. Your research undoubtedly advances our comprehension of necroptosis's complex impact on tumorigenesis.
Author Response
We would like to express our heartfelt appreciation to you for your diligent review and favorable assessment of our manuscript. Your feedback has significantly contributed to the refinement of our research. We remain committed to furthering research in this domain and making substantial contributions to its advancement.
Reviewer 4 Report
Results It is unclear which is the fold change used to dichotomize in high and low mRNA expression
The immunohistochemistry from the tissue database which is the score of proteins in normal tissues and which is the score in tumor
For immunohistochemistry is necessary a higher magnification to evaluate the staining
Methods contain colloquial sentences out of the scope of methods that should describe only the methodology such as at lane 113 £After the preliminary filter and validation, this study further explored the independent-113 ent prognostic value and clinical expression characteristics of the NRGs prognostic OR. In order to identify
Some errors in the text often there is a point and then the sentence start with And some sentence is too confidential such as What’s more, in methods section lane 108
Results It is unclear which is the fold change used to dichotomize in high and low mRNA expression
The immunohistochemistry from the tissue database which is the score of proteins in normal tissues and which is the score in tumor
For immunohistochemistry is necessary a higher magnification to evaluate the staining
Methods contain colloquial sentences out of the scope of methods that should describe only the methodology such as at lane 113 £After the preliminary filter and validation, this study further explored the independent-113 ent prognostic value and clinical expression characteristics of the NRGs prognostic OR. In order to identify
Some errors in the text often there is a point and then the sentence start with And some sentence is too confidential such as What’s more, in methods section lane 108
Reviewer 5 Report
Huang and coworkers present the manuscript entitled “Prognostic Value of Necroptosis-Related Genes Signature in Oral Squamous Cell Carcinoma “. This is another bioinformatics paper with no functional validation which drastically limit the value of the findings.
Authors must validate the signature using they own tumor samples from Chinese population at mRNA and proteins level. Moreover, basic functional analysis of at least one necroptosis gene must be performed to provide novel data on oral carcinoma development and progression. If not his study, although well conducted , it’s limited to bioinformatic predictions, missing biological and functional validation.
Minor editing of English language required.
Round 2
Reviewer 1 Report
Thanks for your revisions. They are sufficient.
Reviewer 2 Report
The authors address each point clearly, and as a result, the quality of the manuscript has improved. I recommend the manuscript for publication.
Reviewer 5 Report
Authors have replied all my concerns, thus I suggest to accept the manuscript for publication in its actual form.
Moderate editing of English language required.